# Chemical Profile, In Vitro Biological Activity and Comparison of Essential Oils from Fresh and Dried Flowers of *Lavandula angustifolia* L.

**DOI:** 10.3390/molecules26175317

**Published:** 2021-09-01

**Authors:** Claudio Caprari, Francesca Fantasma, Fabio Divino, Antonio Bucci, Maria Iorizzi, Gino Naclerio, Giancarlo Ranalli, Gabriella Saviano

**Affiliations:** Department of Bioscience and Territory, University of Molise, C.da Fonte Lappone snc, 86090 Pesche, IS, Italy; claudio.caprari@unimol.it (C.C.); fantasma@unimol.it (F.F.); fabio.divino@unimol.it (F.D.); antonio.bucci@unimol.it (A.B.); iorizzi@unimol.it (M.I.); naclerio@unimol.it (G.N.)

**Keywords:** *Lavandula angustifolia* L., GC-MS analysis, essential oil, antibacterial activity, antifungal activity, antioxidant activity, biodeteriogen control

## Abstract

The chemical composition of essential oils (EOs) from dried and fresh flowers of *Lavandula angustifolia* L. (lavender), named *LA 2019* and *LA 2020,* respectively, grown in central Italy was analyzed and compared by GC and GC-MS. For both samples, 61 compounds were identified, corresponding to 97.9% and 98.1% of the total essential oils. Explorative data analysis, performed to compare the statistical composition of the samples, resulted in a high level of global similarity (around 93%). The compositions of both samples were characterized by 10 major compounds, with a predominance of Linalool (35.3–36.0%), Borneol (15.6–19.4%) and 1,8-Cineole (11.0–9.0%). The in vitro antibacterial activity assay by disk diffusion tests against *Bacillus subtilis* PY79 and *Escherichia coli* DH5α showed inhibition of growth in both indicator strains. In addition, plate counts revealed a bactericidal effect on *E. coli,* which was particularly noticeable when using oil from the fresh lavender flowers at the highest concentrations. An in vitro antifungal assay showed that the EOs inhibited the growth of *Sclerotium rolfsii*, a phytopathogenic fungus that causes post-harvest diseases in many fruits and vegetables. The antioxidant activity was also assessed using the ABTS free radical scavenging assay, which showed a different antioxidant activity in both EOs. In addition, the potential application of EOs as a green method to control biodeterioration phenomena on an artistic wood painting (XIX century) was evaluated.

## 1. Introduction

Among the most investigated natural sources of secondary metabolites, aromatic plants play a fundamental role due to their essential oils (EOs) [1,2,3] and, in the last three decades, there has been a growing interest in their potential use. EOs are variable and complex mixtures of volatile compounds used in the food industry as flavoring agents or for their antibacterial, antifungal and antioxidant properties. The chemical composition of essential oils has a wide range of variability depending on several factors [4]: the geographical area of origin, the environmental and agronomic conditions, the time of harvest, the stage of plant development and the extraction methods [5,6,7,8,9,10]. All of these parameters can modulate different biological activities.

The *Lavandula* genus (family Lamiaceae), a small, fragrant shrub native to the Mediterranean area, is cultivated worldwide for its essential oils, which find applications in various industrial areas, such as perfumes, pharmaceutics and cosmetics; furthermore, lavender essential oil is widely used in the food industry because of its biological properties as well as its attractive aroma [5]. It is effective against the growth of a wide range of microorganisms [11,12] and has antioxidant activity [13].

The *Lavandula* genus consists of over 30 species, which display differences in growth habits and morphological characteristics, including leaf shape, arrangement of the flowers and chemical composition. Only three species are principally cultivated to produce essential oils: fine lavender, the most common (*Lavandula angustifolia*); spike lavender (*Lavandula latifolia*); and lavandin, a hybrid of the two preceding species. The composition of EOs has been extensively investigated [14] due to commercial interest in aromatherapy (relaxant) and in pharmaceutical preparations for its therapeutic effects as a sedative, spasmolytic, antiviral and antibacterial agent [15]. Lavender EO obtained from *L. angustifolia* is the most valuable and the most expensive because its yield is lower than the other EOs [16]. The literature data shows that more than 100 components have been identified; the main active ingredients in *L. angustifolia* flowers’ EO are monoterpenes (Linalool, Linalyl acetate, Lavandulol, Geraniol, Bornyl acetate, Borneol, Terpineol and Eucalyptol (also known as 1,8-Cineole), and Lavandulyl acetate) in different percentages according to the geographic area of origin [12]. Lavender EO from Italian regions shows a high concentration of Linalool (ranging from 35 to 36%), Linalyl acetate (12 to 21%), Camphor (5 to 11%), 1,8-Cineole (3 to 10%) and Borneol (2 to 4%) [12].

EOs from the *Lavandula* genus plants exhibit a broad spectrum of biological activities, and antimicrobial activity has been reported for 65 bacterial strains [17,18,19]. Antimicrobial resistance, and, more specifically, antibiotic resistance in Gram-negative bacteria is a particularly serious public health problem, which poses an urgent need to search for new antibacterial molecules [20]. *Escherichia coli* is one of the most studied species worldwide and comprises non-pathogenic bacteria that may act as commensals and belong to the normal intestinal microbiota of humans and many animals; its pathogenic variants are divided into diarrheagenic and extraintestinal pathogens [21]. In our study, after a preliminary assessment of the antibacterial activity of lavender oil against two strains of *B. subtilis* and *E. coli*, chosen as representatives of Gram-positive and Gram-negative bacteria, the attention was focused on the latter species because significant levels of microbial contamination can occur during the use of cosmetic products, and the possible presence of pathogenic organisms pose a potential risk to health.

The antifungal activity of the essential oil of *L. angustifolia* L. was investigated against 50 clinical isolates of *Candida albicans,* and the EO showed both fungistatic and fungicidal activity [11]. In our study, we evaluated antifungal activity on *Sclerotium rolfsii*; this is a soil-borne polyphagous phytopathogen fungus that causes Southern blight disease and results in significant crop yield losses worldwide, particularly in tropical and subtropical countries [22].

EOs also have antioxidant properties protecting cells against the harmful impact of free radicals. The antioxidant activity of the EO of lavender was shown to inhibit effect fat oxidation reactions and lipid peroxidation in a linoleic acid model system [23]. A DPPH assay or an ABTS assay were also considered as simple and sensitive techniques applicable to most essential oils to assess their antioxidant activity [24].

One of the interesting biological activities concerns the administration of lavender EO through inhalation, which seems effective in reducing anxiety levels by having an effect on the central nervous system. [25].

Recently, for controlling biodeteriogens and biocleaning on artworks related to cultural heritage, a great advantage has reported for the application of biomolecules isolated both from marine organisms (Blue Biotechnology) and from plants organisms (Green Technology) [26,27]. With this in mind, lavender’s essential oils were tested for their potential use in biological control against microbial contamination on artwork surfaces painted on wood.

This study aimed to determine the chemical composition of the EOs of both dried *LA 2019* lavender flowers, stored for one year, and fresh lavender flowers *LA 2020*. Only a few studies describe the changes in the composition of the EOs during the storage of the dry plant material [28,29], and only one study describes the composition changes due to long-term storage of lavender flowers [30].

The purpose of our study was: (i) to compare the chemical composition of dried *LA 2019* and fresh *LA 2020* flowers; (ii) to submit all data to statistical analyses; (iii) to assess the in vitro antibacterial and the antifungal activities against *E. coli, B. subtilis,* and the phytopathogenic fungus *S. rolfsii*, respectively; (iv) to evaluate the antioxidant activities of the EOs by ABTS free radical scavenging assay; (v) to test the EOs in the control of biodeteriogens on artworks related to cultural heritage.

## 2. Results

### 2.1. Essential Oils Yield and Compositions

The hydrodistillation of the flowers of *L. angustifolia*, both *LA 2019* and *LA 2020*, harvested in central Italy (Rosciano, PE) provided an essential oil characterized by a typical smell in a yield of 3.8% and 3.6%, respectively, calculated according to the initial weight of samples used.

Table 1 reports the chemical composition of the EOs and their experimental retention indices compared with the retention indices reported in the literature [31], their percentage compositions and the abbreviations of the different classes of terpenes; the compounds are listed according to their elution on an Rtx^®^-5 Restek capillary column. In this case, 61 compounds were identified, respectively, in samples of both *LA 2019* and *LA 2020* EOs, corresponding to 97.9% and 98.1% of the total area for each sample.

Qualitatively, the composition of both dried and fresh lavender EOs was quite similar, suggesting an almost total conservation of the components (Figure 1). Only two components in low concentrations (Dodecanol and Tetradecenoic acid) were exclusively present in the *LA 2019* sample. Oxygenated monoterpenes represented the most abundant class (84.30% and 87.46%), followed by monoterpenes (3.81% and 3.92%), sesquiterpenes (2.61% and 2.2%) and oxygenated sesquiterpenes (1.69% and 1.27%), as shown in Table 2. Major oxygenated monoterpenes included aliphatic AMO (Linalool 35.3% and 36.03% and Linalyl acetate 3.77% and 2.75%) and bicyclic monoterpenes BMO (1,8-Cineole (or Eucalyptol) 11.0% and 9.0%, Camphor 6.02% and 6.8%, Borneol 15.67% and 19.35%, and Terpinen-4-ol 6.51% and 6.81%). None of the monoterpenes exceeded one percent, while only the sesquiterpene β-Farnesene reached 2.03% and 1.5%, respectively.

### 2.2. Explorative Data Analysis

The data reported in Table 1 are summarized in Figure 2 and Figure 3; the bar plots of the compound percentages, in terms of averages, are reported with the respective standard deviations multiplied by a factor equal to 2 for the *LA 2019* and *LA 2020* EOs, respectively.

In both the *LA 2019* and *LA 2020* sample data, the statistical compositions of the two samples were rather similar and were characterized by the same set of compounds. For instance, by fixing the threshold to 1% for both samples, the compounds with presence percentages greater than such a threshold included the following 10 compounds: 1,8-Cineole, Linalool oxide *cis*, Linalool oxide *trans*, Linalool, Camphor, Borneol, Terpinen-4-ol, Hexyl butyrate, Linalyl acetate and β-Farnesene.

To compare the compounds of the two samples, the following Figure 4 presents a cross-plot of the two sets of percentages with the 2019 data on the x-axis and the 2020 data on the y-axis. The dashed line represents the situation in which a certain compound was present in the same percentages in both samples. The points distant from the line represent compounds observed in rather different percentages with respect to the two samples. In particular, 1,8-Cineole had a level of 11.0% in *LA 2019* and 9.0% in *LA 2020*, and Borneol had levels of 15.7% and 19.4%, respectively.

In order to express the level of similarity between the two samples, the percent model affinity (*PMA*) statistics [32] for the percentages has been calculated:(1)$$PMA=100-0.5\overset{n}{\underset{i=1}{\sum }}\left|p_{19,i}-p_{20,i}\right|,$$
where *n* is the number of compounds considered in both samples, while $p_{19}=\;\left(p_{19,1},\ldots ,p_{19,n}\right)$ and $p_{20}=\;\left(p_{20,1},\ldots ,p_{20,n}\right)$ are the two sets of percentages, *LA 2019* and *LA 2020,* respectively. The use of *PMA* statistics is popular in ecology to detect the level of affinity between two samples in terms of species compositions [32]. In this setting, the *PMA* index was used to compare the *LA 2019* composition with the *LA 2020* one. The value obtained was *PMA* = 93.61%, confirming a high level of global similarity between the two samples.

In Figure 5, the bar plot of the absolute differences that contributed to the calculation of the *PMA* index are reported. The three compounds that showed an absolute difference greater than the threshold of 1% (line in blue) are highlighted by the name: 1,8-Cineole (11.0% vs. 9.0%), Borneol (15.7% vs. 19.4%) and Linalyl (or Linalool) acetate (3.8% vs. 2.8%). Even though these compounds characterized both the compositions of *LA 2019* and *LA 2020* (as stated previously), the differences between the two samples represented potential and specific features that can distinguish the two types of lavender. However, this aspect needs further investigation.

### 2.3. Antibacterial In Vitro Tests

The results of the disk diffusion test are reported in Figure 6. The *LA 2019* and *LA 2020* EOs samples showed a noticeable antibacterial activity against *E. coli* DH5α and *B. subtilis* PY79, with inhibition halos of 20 and 25 mm in diameter, respectively. To determine whether lavender oil had a bactericidal or a bacteriostatic effect on *E. coli* DH5α, various concentrations of lavender oil were added to the indicator strain suspended in PBS buffer. The number of viable bacterial cells was determined by plate counting at various times after the addition of the lavender oil. As shown in Figure 7, the incubation of cells with lavender oil at 25 °C reduced the number of surviving cells, thus suggesting its bactericidal activity. The effect increased with the increase in lavender oil concentration and the incubation time. Finally, the *LA 2020* lavender oil showed a greater bactericidal effect than the *LA 2019* lavender oil.

### 2.4. Antifungal In Vitro Test

*S. rolfsii* has a wide range of hosts, and it is difficult to clear infested soil; it can survive and thrive in a wide range of environmental conditions and in a wide pH range [33]. In order to manage rot infection, the development of resistant varieties and the applications of natural extracts such as natural antifungals are considered a more economical and ecological approach than the application of fungicides. In Italy, *S. rolfsii* is typically present in the Southern regions, but, following hot seasons, infections are occasionally reported in the North as well [34].

The antifungal activity against the growth of *S. rolfsii* phytopathogenic fungus obtained with both *LA 2019* EO and *LA 2020* EO at different concentrations was evaluated in vitro. Figure 8 reports the reduction of fungal growth, expressed as a reduction in mycelium growth. The results show a concentration-dependent trend where the *LA 2020* essential oil was the most effective.

The fungal toxic effect was initially slightly decreased in the *LA 2019* EO; even at an amount of 15 µL, neither extract allowed the in vitro growth of the fungal mycelium. The two samples presented only three compounds with significant differences in terms of quantitative composition. This behavior was probably due to the variation of phytochemicals concentration in the analyzed samples. The positive controls for the antifungal activity were carried out using PDA plates added with mancozeb (mancozeb plus 80 WP, powder, Manica) at final concentrations in the ranges of 0.025–0.05% and 0.2–1%. At lower concentrations (0.025%), there was a reduction (30%) in the mycelial growth of *S. rolfsii.* All other concentrations (0.05% and 0.2–1%) showed a “no growth, mycelium-free” result on the mancozeb PDA plates [34].

### 2.5. Antioxidant Test

The antioxidant properties of *LA 2019* and *LA 2020* were evaluated. Antioxidant activity was determined using an ABTS radical scavenging assay. This assay is based on the use of a solution containing a radical substance (ABTS^•+^), whose absorbance at 734 nm decreases in proportion to the amount of antioxidant compounds added. The positive control used in this research was ascorbic acid. The inhibition concentration, IC_50_, was measured, and the results are listed in Table 3. IC_50_ was calculated by plotting the % of inhibition against the concentration in mg/mL

As shown in Figure 9, the inhibition percentage of the free radical increased with the increase in the concentration of the essential oils of lavender. The free radical scavenging activity of both EOs was observed at a concentration of 0.041 mg/mL to 8.27 mg/mL. The percentage of the inhibition of ABTS from the *LA 2019* EO was slightly higher (99.48%) than that of the *LA 2020* EO (96.88%).

### 2.6. Biodeteriogen Control on Altered Painting

In Figure 10a, the total viable microbial count is reported. On the surface of the paint, the total count of viable bacteria ranged from 3 × 10 to 2 × 10^2^ UFC/plate dish after sampling using the sterile velvet technique; the total viable micromycete (fungi) count was less than 5 × 10 UFC/plate, and there was an absence of yeasts. In Figure 10b, some isolated bacteria yellow pigmented are reported.

After observations, the bacterial strain *Micrococcus luteus* was identified. In Figure 10c,d, the bacterial growth inhibition by halo diameters underline the effectiveness of the antimicrobial activity of the EOs tested. In fact, the *LA 2019* and *LA 2020* EO samples showed remarkable differences. The average diameters of the inhibition halos measuring the antimicrobial response (sensible ≥ 6 mm, resistant < 6 mm) suggest higher efficacy when *LA 2020* was adopted.

## 3. Discussion

Plant metabolites and plant-based pesticides appear to be one of the best alternatives as they have minimal environmental impact and present less hazard to consumers compared to synthetic pesticides. For this purpose, plant extracts and essential oils are ecological, protective, curative and antagonistic to many diseases. Therefore, plant extracts and biological agents can be used as an alternative source to control soil-borne diseases as they are a rich source of bioactive substances [35].

A comparison between *L. angustifolia* EOs described in the literature highlights the influence of geographic region of origin on the composition and the effective activity or specificity against particular microorganisms [6,7,8,9,10]. Previous studies have indicated that lavender EOs from Italy are rich in Linalool (35–36%), Linalyl acetate (12–21%), Camphor (5–11%), 1,8-Cineole (3–10%), Terpinene-4-ol (2–6%), β-Farnesene (1–4%) and Borneol (2–4%) [12].

In both *LA 2019* and *LA 2020,* the composition of the EOs from *L. angustifolia* from the Abruzzo region was mainly characterized by a high Borneol content (15.7% and 19.4%), Linalool (35.3% and 36.0%) and 1,8-Cineole (11.0% and 9.0%), while the concentration of Linalyl acetate (3.77% and 2.75%) was low. In the fresh sample *LA 2020*, a high concentration of Borneol and Linalool was associated with a low concentration of 1,8-Cineole (9%); in the dried sample *LA 2019*, the level of Linalool and Borneol slightly decreased, while the level of 1,8 Cineol (11%) increased.

It should be noted that a high concentration of Borneol (22.6%), associated with a high level of 1,8-Cineole (39.8%) and Camphor (22.1%), has previously been observed only in the EO of *L. angustifolia* obtained from leaves and stem grown in Brazil, which showed anti-inflammatory activity. [36]. The relatively small amount of Linalyl acetate in respect to the literature data can be attributed to partial decomposition during hydrodistillation [13]. Most of these components belong to the alcohol group, and, among them, linalool was demonstrated to be the strongest active ingredient against a wide range of microorganisms [37]. The antibacterial activity of 33 free terpenes commonly found in essential oils was recently investigated [38]. It was demonstrated that Borneol, in both enantiomeric forms, exhibited antimicrobial activity against *B. cereus, E. coli* and *S. aureus* (MIC ranging from 0.03 to 0.25 mg/mL) and *S. typhimurium* (MIC 0.12 to 800 mg/mL). Similar antibacterial activity was found for (±) Linalool (MIC 0.25 mg/mL), while 1,8-Cineole was inactive against all strains tested [38].

It is known that the synergistic or antagonistic effects of the chemical components in the EOs can contribute to the difference in the biological activities [39]; however, as both our *LA 2019* and *LA 2020* samples contained a large amount of Linalool and Borneol, we can assume that the antibacterial and antifungal activity was in part influenced by the concentration of these components. Indeed, the higher concentration of Linalool (36.0%) and Borneol (19.4%) in the fresh lavender sample (*LA 2020*) resulted in higher antimicrobial activity than the same activities observed in the dried lavender sample (*LA2019*). Borneol in elevated concentrations may be considered as marker component of *Lavandula* genus from the Abruzzo region, and this is probably related to the local microhabitat in which *L. angustifolia* is cultivated (see Plant Material). This particular chemical composition is reported here for the first time.

The composition of the EOs may explain the different values of IC_50_ in the antioxidant test. The high content of monoterpenes and monoterpenoids may be responsible for the antioxidant activity. The higher antioxidant activity of *LA 2019* EO could be related to the synergistic property associated with the high content of sesquiterpenes and their derivatives (Table 2) [40].

The preliminary results of the antibacterial assays on *B. subtilis* PY79 and, specifically, on *E. coli* DH5α cells are encouraging and set the stage for further research focused on clinically relevant bacterial species. Antimicrobial resistance in bacteria, especially resistance in Gram-negative bacteria, is concerning. This complex and relatively old phenomenon is the consequence of the vertiginous decrease in research and the development of new antibacterial compounds and the appearance and spread of resistant or even multidrug-resistant bacteria [20]. Accordingly, one of the priorities of research should be the search for new antibacterial molecules since the therapeutic arsenal is shrinking and there is a lack of new molecules with new chemical structures or new mechanisms of action to fight the Gram-negative bacteria that currently pose the greatest threat to human health. Antibacterial activity assays against *E. coli* DH5α revealed a bactericidal effect that was particularly noticeable when using oil from fresh lavender flowers at the highest concentrations (30 μL and 150 μL). It is likely that the stronger impact observed after testing the *LA 2020* EOs at the highest concentrations could also be due to those compounds characterizing potential and specific features of the investigated oil samples.

The metabolic functions of microorganisms can be altered by gaseous contact and exposure to essential oils. One study reports that vapor levels of 0.1–0.9 mg/L in air may suppress the growth of the bacterial pathogens responsible for respiratory infections [41]. Several EOs, such as those from cinnamon bark, lemongrass and thyme, have been shown to be the most effective in this field. Antimicrobial and antiseptic properties and antiviral activity have been studied in eucalyptus oils and tea tree oil (*Melaleuca alternifolia*) [42,43,44], and their use as disinfectants for indoor air quality after adequate development of air technologies has been suggested [45,46].

Antifungal activity was performed in vitro with the pathogen *S. rolfsii* Sacc. (teleomorph *Athelia rolfsii* (Curzi) C. C. Tu and Kimbr.), which is considered one of the most destructive pathogens worldwide, characterized by wide host range comprising more than 500 plant species from 100 families, including tomatoes, potatoes, chili peppers, carrots, cabbage, sweet potatoes, common beans and ground nuts [47,48]. *S. rolfsii*, a polyphagous pathogenic fungus and infects a very large number of plant species of agricultural interest with a mortality ranging from 10 to 100% [49]. Due to the abundant production of sclerotia that can persist in the soil for years, the management of the pathogen can be achieved with various fungicides, soil fumigants (methyl bromide), plant extracts and biological agents [35].

Essentially, the *LA 2019* and *LA 2020* Eos were both active against the fungus, but the fungal toxic effect was slightly greater with *LA 2020* than *LA 2019*. Our results suggest that the greater antifungal activity of *LA 2020* can be related to the higher concentration of Borneol, and the synergistic or antagonistic effects of other compounds, as reported, can amplify its antifungal activity [39]. Some mechanisms have been proposed, such as the possibility that terpenes can increase the concentration of lipid peroxides and cause cell death [50], or that they can act on the hyphae of the mycelium, inducing the release of components from the cytoplasm and hence the death of the mycelium [51].

A current and relevant problem in the field of cultural heritage is the conservation of wooden paintings, which are potentially attacked by microorganisms and pests over the years. Preliminary results need confirmation from extensive antibacterial tests on both isolated microbes and the complex microbial community of the altered artwork.

In the case of antibacterial and antifungal testing, we need an in-depth study that includes more strains of both bacteria and fungi (both resistant and sensitive strains where possible) to definitively conclude the antibacterial and antifungal activities of these EOs. Moreover, further research will explore the cytotoxicity of the EOs to support the final correct operative decisions.

These indications, if functional and supported by solid statistical analyses, could be useful to develop a new organic green strategy for the recovery of altered works of art and as an alternative to the use of biocides and toxic compounds.

From these perspectives, the use of *L. angustifolia* EOs as a natural essence in a confined space, such as a bag containing the work of art needing restoration, for an adequate amount of time, could be a suitable technical solution.

## 4. Materials and Methods

### 4.1. Plant Materials

*L. angustifolia* flowers were collected in September 2019 and September 2020, always in the early hours of the morning by the same staff. The flowers were harvested from the same shrubs in the same area during the balsamic period, at Villa Vanda Farm (42°20′59.83″ N, 14°01′54.88″ E) at an altitude of about 150 m and at a distance of 20 km from the Adriatic Sea. The flowers are grown in parallel rows and in association with a centuries-old olive groves managed according to the rules of organic farming in an area called “Oasi orientale di Villa Badessa”, Rosciano (Pe, Abruzzo Region, Italy).

The soil has a predominantly loamy-clayey texture with no summer irrigation. The plant was identified by Dr. Fortini, and a voucher specimen was deposited in the Herbarium of DiBT, University of Molise. The flowers harvested in 2019 were well preserved for one year in the dark at room temperature (21 °C) and were then submitted to hydrodistillation to give the EO (named dried *LA 2019*). The flowers harvested in 2020 were immediately used for the hydrodistillation of the EO (named fresh *LA 2020*). The composition of both samples was analyzed and compared.

### 4.2. Essential Oil Isolation

Defined amounts of flowers (200 g) both of dried (2019) and fresh (2020) *L. angustifolia* were hand selected and cleaned, and then separately subjected to hydrodistillation for 3 h according to the standard procedure described in the Council of Europe [52]. The essential oils were dried over anhydrous sodium sulfate to remove traces of water and then stored in dark vials at 4 °C prior to gas chromatography-mass spectrometry (GC-MS) analysis.

### 4.3. GC-FID Analysis

The characterization of both essential oils samples was determined with a gas chromatography system GC 86.10 Expander (Dani) equipped with an FID detector, Rtx^®^-5 Restek capillary column (30 m × 0.25 mm i.d., 0.25 μm film thickness) (diphenyl-dimethyl polysiloxane), a spilt/splitless injector heated to 250 °C and a flame ionization detector (FID) heated to 280 °C. The column temperature was maintained at 40 °C for 5 min, then programmed to increase to 250 °C at a rate of 3 °C/min and held, using an isothermal process, for 10 min; the carrier gas was He (1.0 mL/min); 1 μL of each sample was dissolved in *n*-exane (1:500 *n*-exane solution) and injected. The experiment was repeated three times.

### 4.4. GC/MS Analysis

The GC-MS analyses were performed on a Trace GC Ultra (Thermo Fisher Scientific) gas chromatography instrument equipped with a Rtx^®^-5 Restek capillary column (30 m × 0.25 mm i.d., 0.25 µm film thickness) and coupled with an ion-trap (IT) mass spectrometry (MS) detector Polaris Q (Thermo Fisher Scientific, Waltham, MA, USA).

A Programmed Temperature Vaporizer (PTV) injector and a PC with a chromatography station Xcalibur (Thermo Fisher Scientific) was used. The ionization voltage was 70 eV; the source temperature was 250 °C; full scan acquisition in positive chemical ionization was from *m*/*z* 40 up to 400 a.m.u. at 0.43 scan s^−1^. The GC conditions were the same as those described above for the gas chromatography (GC-FID) analysis.

### 4.5. Identification of Essential Oil Components

The identification of the essential oil components was based on the comparison of their Kovats retention indices (RIs) and RI (linear retention indices), determined in relation to the tR values of a homologous series of *n*-alkanes (C8–C20) injected under the same operating conditions as those described in the literature [53,54].

The MS fragmentation patterns of a single compound were those from the NIST 02, Adams and Wiley 275 mass spectral libraries [55,56]. The relative contents (%) of the sample components were computed as the average of the GC peak areas obtained in triplicate without any corrections [57].

### 4.6. Statistical Analysis

The explorative data analysis was performed using R software, available as free software under the terms of the Free Software Foundation’s GNU General Public License in source code form.

### 4.7. Antibacterial Activity Assays against B. subtilis PY79 and E. coli DH5α

As a preliminary step, the antibacterial properties of lavender oil against *B. subtilis* PY79 and *E. coli* DH5α cells were determined using the paper disk diffusion method. Sterile paper disks 6 mm in diameter were placed onto Luria–Bertani (LB) agar plates, previously inoculated with 150 μL of *B. subtilis* PY79 [58] or *E. coli* DH5α [59] cultures grown in LB broth at 37 °C, with agitation at 120 rpm until an optical density (OD) at 600 nm of 1.0 was achieved. Then, the disks were impregnated with 10 μL of the essential oil samples *LA 2019* and *LA 2020*. Disks with PBS 1X, LB broth and mineral oil were used as negative controls, whereas a disk containing amoxicillin (20 μg) + clavulanic acid (10 μg) was used as a positive control. The plates were incubated at 37 °C for 24 h, and the diameter of inhibitory zones was measured. The experiment was repeated three times. Afterwards, in order to assess if the EOs had a bactericidal or a bacteriostatic effect on *E. coli* DH5α, various volumes (3, 30 and 150 μL) of lavender oil were added to ≈10^7^ cells re-suspended in PBS 1X, for a final volume of 300 μL. Viable counts were performed at different times from the exposure (5, 30 and 120 min). The experiment was repeated three times.

### 4.8. Antifungal Activity Assay

The mold strain of *S. rolfsii,* previously identified and characterized [34], was used in this study. Pure essential oils from the two *LA 2019* and *LA 2020* samples were dissolved in a final volume of 200 µL in ethanol and then added to 19 mL PDA plates to obtain the different final concentrations. Mycelial plugs (4 mm in diameter) from the edges of the *S. rolfsii* 5-day culture were incubated in the center of each PDA plate (90 mm diameter). The fungal cultures were incubated in the dark at 27 °C and 70% relative humidity (RH) for 3 days. The tests were conducted in triplicate. The antifungal activity was determined by measuring the diameter (in mm) of radial growth. The control growth was carried out on PDA plates prepared as described above, but without the EO samples. The positive controls for the antifungal activity were carried out using PDA plates added with mancozeb (Mancozeb plus 80 WP, powder, Manica SpA, Rovereto, Italy) at final concentrations in the range of 0.025–0.05% and 0.2–1%. The growth of the *S. rolfsii* mycelia control was monitored on the PDA plates at 27 °C and 70% relative humidity (RH) for 3 days.

### 4.9. Antioxidant Activity

The ABTS method was used based on the ability of antioxidant molecules to quench the long-lived ABTS^•+^. The ABTS^•+^ was generated by peroxydisulfate oxidation 2,2ʹ azinobis (3-ethylbenzothiazoline-6-sulfonic acid) following the reported method with minor changes [39]. Briefly, the ABTS^•+^ solution was prepared by mixing 5 mL of a 7 mM solution of ABTS in water with 88 μL of a 140 mM solution of potassium persulfate. The solution was stored at room temperature 16 h in the dark. The working solution was prepared by adding methanol until the absorbance was large (0.70 ± 0.001) at 734 nm. The samples were prepared by mixing different volumes of EOs (ranging from 0.05 to 20 μL) in 100 μL of 70% final methanol solution. The control contained only 70% methanol in water. A volume of 1 mL of working solution was mixed with 100 μL of samples and incubated for 4 min at room temperature. After incubation, the absorbance of the reaction mixture was measured at 734 nm. The experiment was performed in triplicate. The ability of the essential oil to scavenge ABTS radicals was calculated as % inhibition by the following Equation: % inhibition = [(Abs control − Abs sample)/(Abs control)] × 100, where Abs control is the absorbance of the ABTS^•+^ + methanol; Abs sample is the absorbance of ABTS^•+^ + EO. Ascorbic acid was used as a standard.

### 4.10. Case Study: A Painting on Wood Dated from the XIX Century

A historical wood painting was chosen for the ex situ study “*Madonna con Bambino*”, originally located on the inner wall of the *S. Maria del Lago* church at Pesche (Isernia, Italy), Figure 11a. The artwork (140 cm × 55 cm × 3.5 cm) painted by an anonymous painter was submitted to microbiological sampling on the basis of aesthetical alterations on the surface of the paint after preliminary observation by R. Sartorio Restorer (Uraniarst, Isernia, Italy), commissioned by the Bishop in charge at the Cathedral-Bishop’s Curia, Isernia. The water painting was irregularly affected by very small lacunas, visible even to the naked eye, only in some areas on the paint’s surface.

Microbiological analyses were carried out as follows: (i) sampling by soft and non-destructive techniques using sterile cotton swabs and sterile velvet tampons (Figure 11b); (ii) the inoculation of the samples on sterile Petri dishes containing cultural solid media for the growth of total aerobic bacteria (Standard Plate Count Agar, Difco; incubation at 37 °C for 48–72 h); for the total micromycetes (fungi and yeasts, Peptone Destrosio Agar Rosa Bengala, Difco), the incubation was at 28 °C for 48 h.

### 4.11. Antimicrobial Activity on Paint

The antimicrobial activities (antagonistic properties) of the isolated bacteria were evaluated in vitro using the EO diffusion agarized nutrient method. Only the yellow-pigmented bacterial strains typically representative of biological particulates both in the air environment and the surface of the potential biodeteriorated paint were identified. The tests included morphological properties under microscopic observations and physiological and biochemical tests by a Gram reaction and Api-Systems (Biomerieux Italia, Firenze) according to Bergey’s Manual [60].

For the assay against yellow-pigmented isolated bacteria, the *L. angustifolia* EOs were tested using both agar wells and disk diffusion on Petri dishes containing Plate Count Agar (PCA, Difco) and inoculation of selected strains. The suspensions containing exponentially growing bacteria after an overnight broth culture, at approximately 10^8^ CFU / mL, corresponding to an O.D. 560 nm of approximately 1.6, were used [61,62].

Paper disks with EO microdilution (10 μL/disk) were carried out in triplicate. As a control, a sterile physiological solution instead of EO dilutions were adopted. The plates were incubated at 37 °C for 48–72 h, and the halos of inhibition around the disks were measured (in mm).

## 5. Conclusions

In the present study, a relationship between the composition of fresh and dried EOs of flowers extracted from *L. angustifolia* and the antimicrobial and antioxidant activity was observed. The two samples (*LA 2019* and *LA 2020*) were qualitatively similar and contained Borneol and Linalool in high concentrations. This is the first time that an EO from *L. angustifolia* with a high Borneol content has been described in Italy. The differences observed between the two samples (dried and fresh lavender) concerned only the concentration of Borneol, Linalool and 1,8-Cineole, suggesting an almost total conservation of the components during the drying process. In our experimental conditions, we presumed that the antibacterial and antifungal activity was partially influenced by the concentration of these components.

The Borneol-rich oils harvested in central Italy were found to have antibacterial activity against both *B. subtilis* PY79 and *E. coli* DH5α, with a significant bactericidal effect on the latter as well as an antifungal effect on the phytopathogen fungus *S. rolfisii*. These activities are associated with a scavenging capacity; therefore, our results suggest that the EO can be used as potential biopreservatives in several fields by reducing and inhibiting pathogenic bacteria. Their antioxidant activity makes *L. angustifolia* EOs a favorable candidate for decreasing the level of microbial contamination not only in different categories such as cosmetic products (lipstick, beauty creams, etc.) but also in potential “natural” alternatives to synthetic antioxidants/bactericides.

The interesting antimicrobial activities, if confirmed with further studies on the biodeteriogen microflora associated with the artworks, could be useful in establishing a new green biological strategy for the recovery of altered artworks and as an alternative to the use of biocides and toxic compounds.

## Figures and Tables

**Figure 1 molecules-26-05317-f001:**
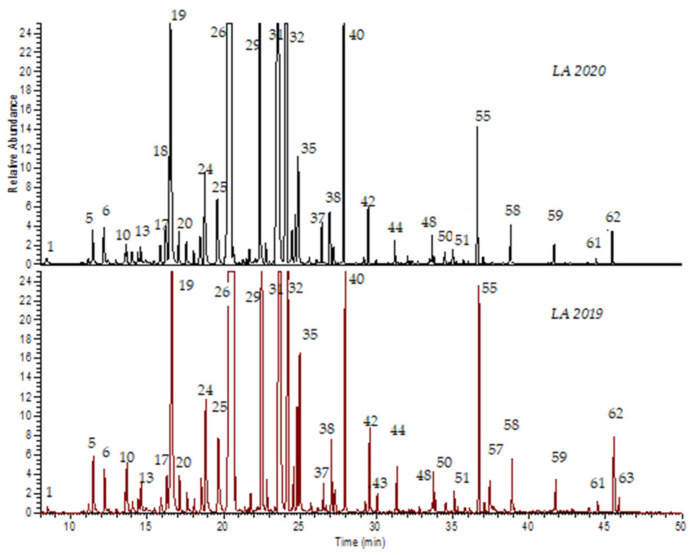
The GC-MS TIC chromatograms of the *LA 2019* and *LA 2020* samples. Numbers refer to those reported in Table 1.

**Figure 2 molecules-26-05317-f002:**
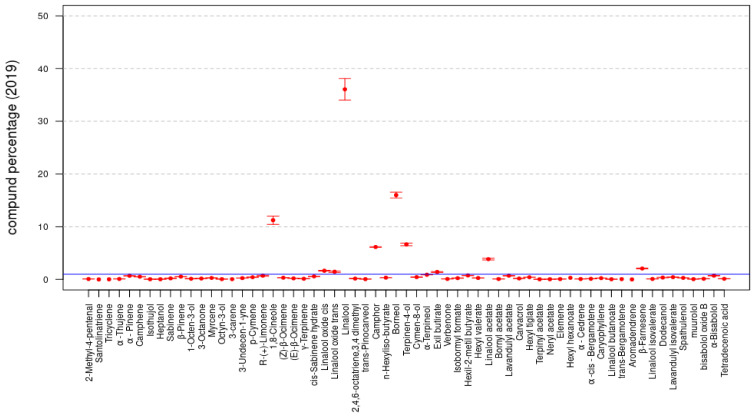
Compound percentages, mean ± SD, *LA 2019* EO.

**Figure 3 molecules-26-05317-f003:**
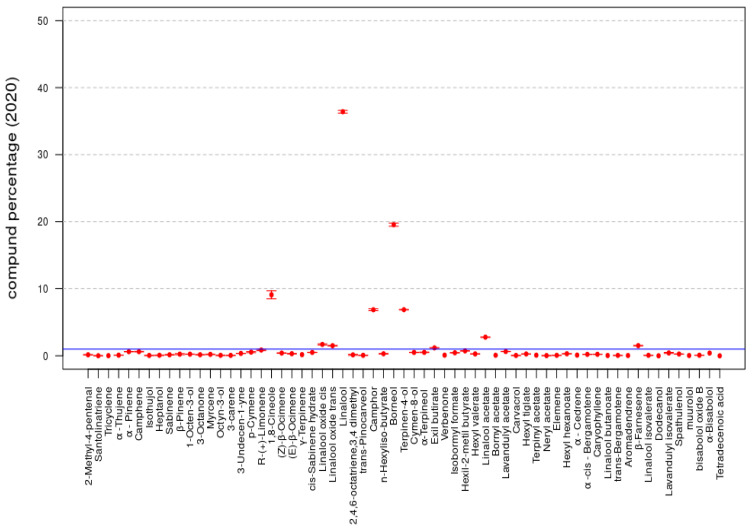
Compound percentages, mean ± SD, *LA 2020* EO.

**Figure 4 molecules-26-05317-f004:**
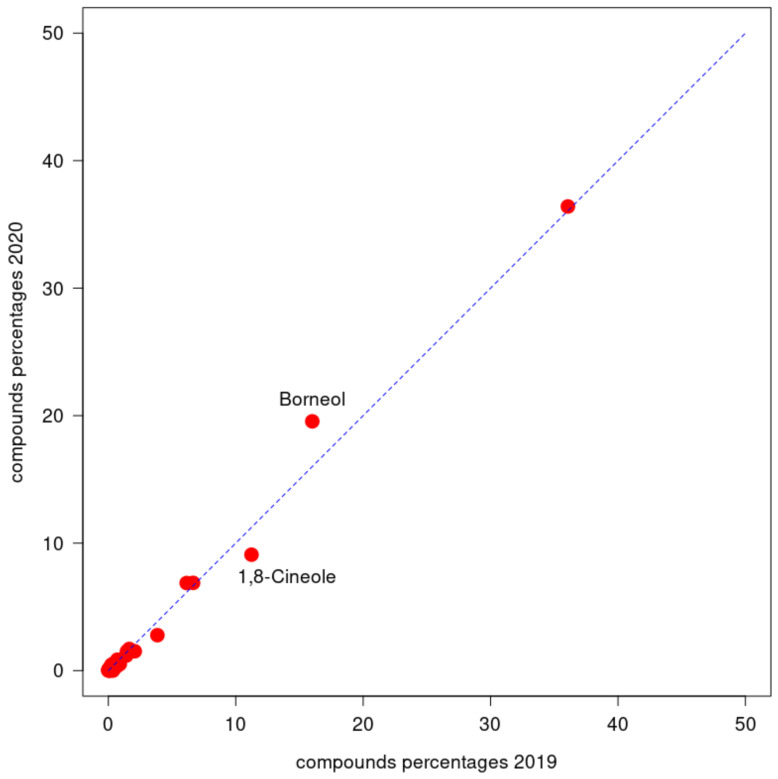
Cross plot of *LA 2019* percentages of components vs. *LA 2020* percentages of components.

**Figure 5 molecules-26-05317-f005:**
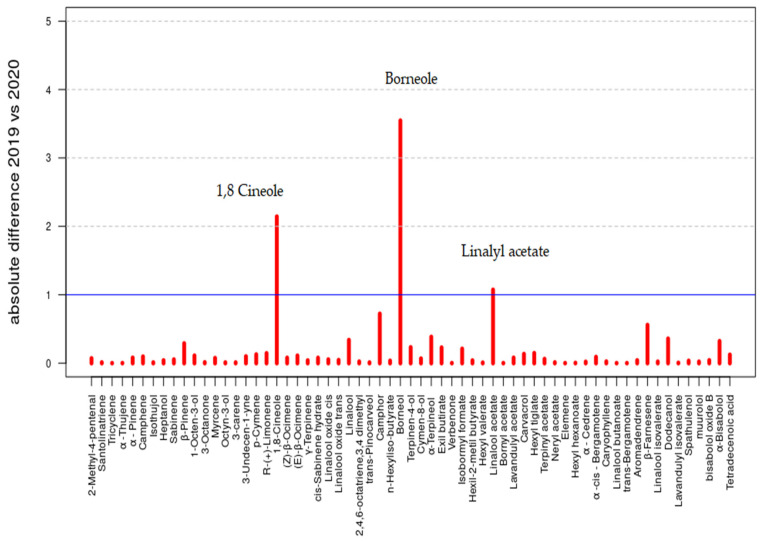
Bar plot of the absolute differences between *LA 2019* and *LA 2020* percentages.

**Figure 6 molecules-26-05317-f006:**
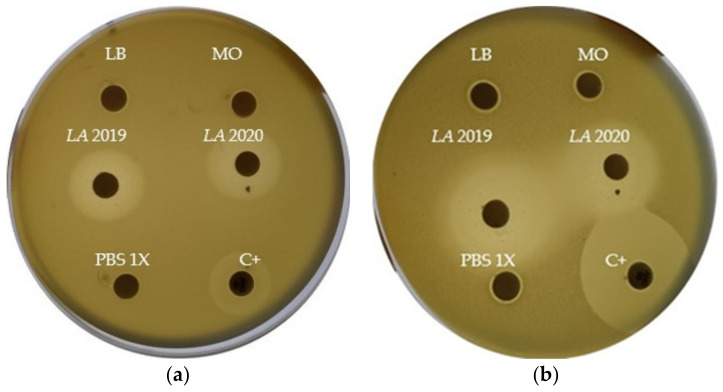
Zones of growth inhibition (halos) of *E. coli* DH5α (**a**) and *B. subtilis* PY79 (**b**) in the presence of 10 μL of *LA 2019* and *LA 2020* EOs. We used 10 μL of LB broth (LB), 10 μL of mineral oil (MO) and 10 μL of PBS 1X as negative controls, and amoxicillin (20 μg) + clavulanic acid (10 μg) as positive control (C+).

**Figure 7 molecules-26-05317-f007:**
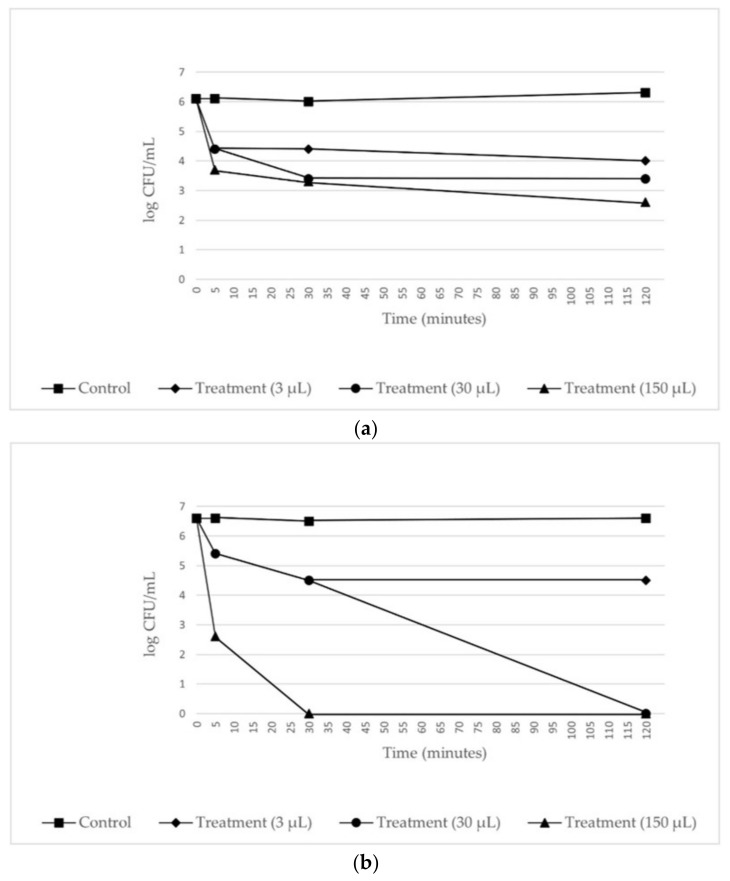
Effects of different concentrations of *LA 2019* EO (**a**) and *LA 2020* EO (**b**) on the growth of the indicator strain *E. coli* DH5α. Exponentially growing cells of the indicator strain were incubated with the lavender oils at 25 °C, and samples were tested for growth after various incubation times.

**Figure 8 molecules-26-05317-f008:**
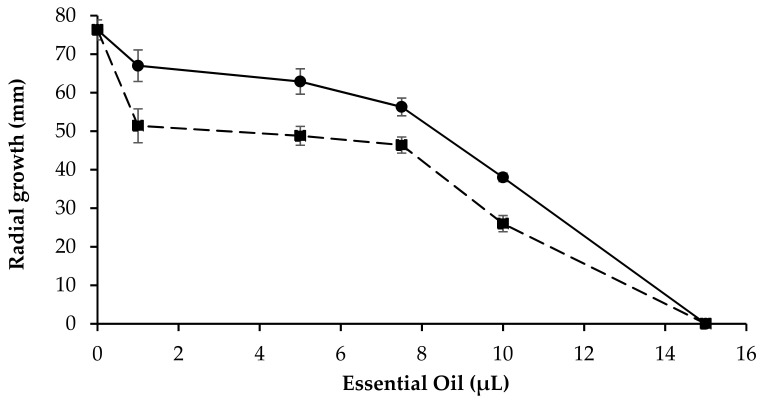
Toxic effect with different concentrations of the *LA 2019* (●) and *LA 2020* (▪) pure EOs of *L. augustifolia* on the radial growth of *S. rolfsii*. Tests were conducted in triplicate ± SE.

**Figure 9 molecules-26-05317-f009:**
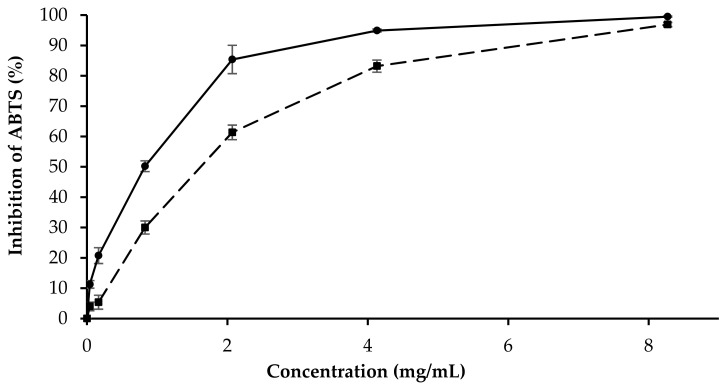
Scavenging effect of the two different essential oils of *L. augustifolia*, *LA 2019* (●) and *LA 2020* (▪) on ABTS assays at different concentrations ranging from 0.041 mg/mL to 8.5 mg/mL. Data are expressed as mean values ± SE (*n* = 3).

**Figure 10 molecules-26-05317-f010:**
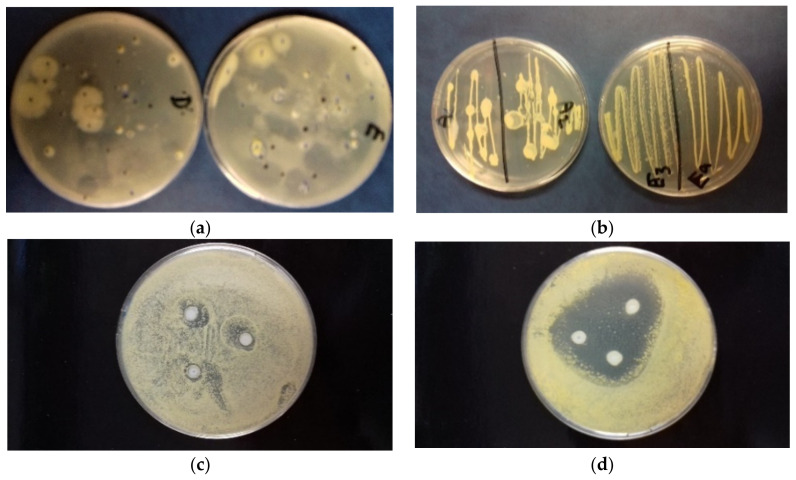
(**a**) Microbial growth on agarized cultural medium from paint surface sampled using the sterile velvet technique. (**b**) Isolation of viable and yellow pigmented bacterial strains, *Micrococcus luteus.* (**c**) Antibacterial in vitro test carried out with LA 2019 EO and the absence of antimicrobial effects (halo < 6 mm). (**d**) Antibacterial in vitro test carried out with LA 2020 EO and the presence of antimicrobial effects (halo > 6 mm).

**Figure 11 molecules-26-05317-f011:**
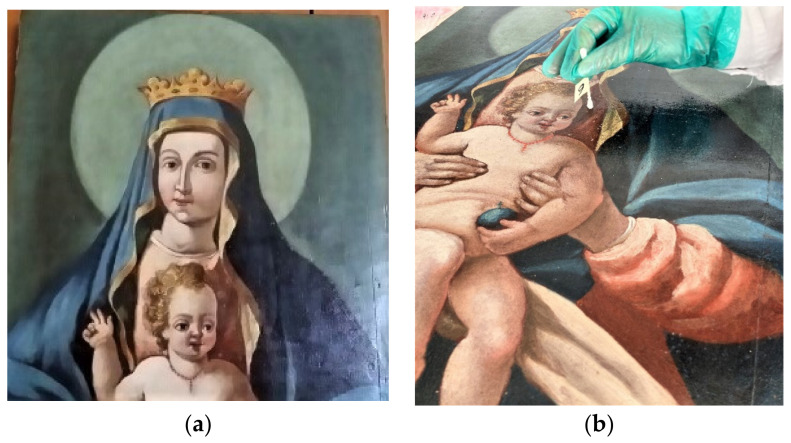
(**a**) Historical wood paintings, “*Madonna con Bambino*”, originally located in the *Santa Maria del Lago* church, chosen for the ex situ study. (**b**) Sampling phase used sterile cotton swabs for microbiological analyses, with permission of the Bishop in charge of the Cathedral-Bishop’s Curia, Isernia.

**Table 1 molecules-26-05317-t001:** Chemical composition of the essential oils (EOs) extracted from the flowers of *L. angustifolia*.

No.	Compound	Exp. RI	Ref. RI	*LA 2019* Area % ± SD	*LA 2020* Area % ± SD	Abbr.
1	2-Methyl-4-pentenal	-	732	0.08 ± 0.01	0.15 ± 0.01	OT
2	Santolinatriene	918	908	t	0.01 ± 0.01	AM
3	Tricyclene	922	926	0.02 ± 0.00	0.02 ± 0.00	BM
4	α-Thujene	929	930	0.09 ± 0.01	0.09 ± 0.01	BM
5	α-Pinene	936	939	0.68 ± 0.02	0.61 ± 0.01	BM
6	Camphene	950	954	0.51 ± 0.02	0.61 ± 0.02	BM
7	Isothujol	955	960	0.04 ± 0.01	0.05 ± 0.01	BMO
8	Heptanol	965	996	0.04 ± 0.01	0.08 ±0.02	OT
9	Sabinene	976	975	0.20 ± 0.03	0.15 ± 0.02	BM
10	β-Pinene	977	979	0.52 ± 0.01	0.24 ± 0.04	BM
11	1-Octen-3-ol	984	979	0.13 ± 0.02	0.24 ± 0.01	OT
12	3-Octanone	990	983	0.17 ± 0.01	0.16 ± 0.01	OT
13	Myrcene	993	990	0.28 ± 0.02	0.21 ± 0.02	AM
14	Octyn-3-ol	998	-	0.06 ± 0.01	0.07 ± 0.02	OT
15	3-carene	1009	1011	0.05 ± 0.00	0.06 ± 0.01	BM
16	3-Undecen-1-yne	1018	-	0.25 ± 0.01	0.35 ± 0.01	AM
17	*p*-Cymene	1026	1024	0.41 ± 0.03	0.54 ± 0.04	MM
18	*R*-(+)-Limonene	1031	1029	0.70 ± 0.05	0.85 ± 0.05	MM
19	1,8-Cineole	1034	1031	11.0 ± 0.4	9.0 ± 0.3	BMO
20	(*Z*)-β-Ocimene	1043	1037	0.32 ± 0.02	0.40 ± 0.02	AM
21	(*E*)-β-Ocimene	1053	1050	0.20 ± 0.01	0.31 ± 0.02	AM
22	γ-Terpinene	1062	1059	0.13 ± 0.01	0.17 ± 0.00	MM
23	*cis*-Sabinene hydrate	1069	1070	0.57 ± 0.01	0.50 ± 0.01	BMO
24	Linalool oxide *cis*	1075	1072	1.61 ± 0.04	1.68 ± 0.04	AMO
25	Linalool oxide *trans*	1089	1086	1.43 ± 0.06	1.49 ± 0.02	AMO
26	Linalool	1105	1096	35.3 ± 1.0	36.0 ± 0.1	AMO
27	2,4,6-octatriene,3,4 dimethyl	1133	1132	0.16 ± 0.02	0.14 ± 0.04	AM
28	*trans*-Pinocarveol	1142	1139	0.06 ± 0.01	0.07 ± 0.01	BMO
29	Camphor	1148	1146	6.02 ± 0.04	6.80 ± 0.07	BMO
30	*n*-Hexyliso-butyrate	1154	1150	0.33 ± 0.02	0.30 ± 0.02	AMO
31	Borneol	1170	1169	15.7 ± 0.3	19.4 ± 0.1	BMO
32	Terpinen-4-ol	1182	1177	6.5 ± 0.1	6.81 ± 0.02	BMO
33	Cymen-8-ol	1189	1182	0.43 ± 0.03	0.50 ± 0.01	MMO
34	α-Terpineol	1193	1188	0.88 ± 0.01	0.51 ± 0.02	MMO
35	Hexyl butyrate	1196	1192	1.38 ± 0.04	1.17 ± 0.02	OT
36	Verbenone	1212	1205	0.09 ± 0.01	0.09 ± 0.00	BMO
37	Isobormyl formate	1231	1239	0.24 ± 0.02	0.45 ± 0.01	BMO
38	Hexyl-2-metil butyrate	1242	1236	0.75 ± 0.02	0.72 ± 0.01	OT
39	Hexyl valerate	1247	1244	0.27 ± 0.01	0.28 ± 0.01	OT
40	Linalyl acetate	1262	1257	3.77 ± 0.08	2.75 ± 0.02	AMO
41	Bornyl acetate	1288	1288	0.08 ± 0.01	0.08 ± 0.00	BMO
42	Lavandulyl acetate	1295	1290	0.70 ± 0.02	0.63 ± 0.01	AMO
43	Carvacrol	1306	1299	0.18 ± 0.01	0.05 ± 0.01	MMO
44	Hexyl tiglate	1336	1332	0.41 ± 0.01	0.27 ± 0.01	OT
45	Terpinyl acetate	1354	1349	0.03 ± 0.01	0.09 ± 0.00	OT
46	Neryl acetate	1369	1361	0.04 ± 0.01	0.03 ± 0.01	AMO
47	Elemene	1388	1390	0.07 ± 0.01	0.07 ± 0.01	MS
48	Hexyl hexanoate	1390	1383	0.30 ± 0.00	0.30 ± 0.01	AMO
49	α-Cedrene	1392	1411	0.08 ± 0.01	0.10 ± 0.00	BS
50	α-*cis*-Bergamotene	1408	1412	0.12 ± 0.01	0.21 ± 0.01	MS
51	Caryophyllene	1422	1419	0.24 ± 0.01	0.22 ± 0.01	BS
52	Linalool butanoate	1428	1423	0.04 ± 0.01	0.04 ± 0.00	AMO
53	*trans*-Bergamotene	1439	1434	0.06 ± 0.00	0.06 ± 0.01	MS
54	Aromadendrene	1444	1441	t	0.04 ± 0.00	BS
55	β-Farnesene	1462	1456	2.03 ± 0.04	1.50 ± 0.01	AS
56	Linalool isovalerate	1470	1468	0.09 ± 0.01	0.07 ± 0.01	ASO
57	Dodecanol	1478	1470	0.35 ± 0.03	-	OT
58	Lavandulyl isovalerate	1514	1509	0.42 ± 0.03	0.42 ± 0.04	ASO
59	Spathulenol	1589	1578	0.29 ± 0.01	0.26 ± 0.01	BSO
60	Muurolol	1646	1646	0.06 ± 0.01	0.04 ± 0.00	BSO
61	Bisabolol oxide B	1661	1658	0.12 ± 0.01	0.08 ± 0.01	BSO
62	α-Bisabolol	1688	1685	0.71 ± 0.02	0.40 ± 0.00	MSO
63	Tetradecenoic acid	1696	-	0.12 ± 0.01	-	OT

Abbreviations: AM: aliphatic monoterpenes; MM: monocyclic monoterpenes; BM: bi-and tricyclic monoterpenes; AMO: aliphatic monoterpenoids; MMO: monocyclic monoterpenoids; BMO: bi-and tricyclic monoterpenoids; AS: aliphatic sesquiterpenes; MS: monocyclic sesquiterpenes; BS: bi- and tricyclic sesquiterpenes; ASO: aliphatic sesquiterpenoids; MSO: monocyclic sesquiterpenoids; BSO: bi- and tricyclic sesquiterpenoids, OT: others. SD: standard deviation; Exp. RI: experimental retention index; Ref. RI: literature data; t: traces.

**Table 2 molecules-26-05317-t002:** List of terpenes in the lavender oils.

	Abbreviation	*LA 2019* and Area %	*LA 2020* and Area %
Aliphatic monoterpenes Monocyclic monoterpenes Bi–and tricyclic monoterpenes	AM MM BM	1.2 1.24 2.07	1.42 1.56 1.79
**Monoterpenes**	**M**	**3.81**	**3.92**
Aliphatic monoterpenoids Monocyclic monoterpenoids Bi–and tricyclic monoterpenoids	AMO MMO BMO	43.54 1.48 40.28	43.2 1.06 43.2
**Monoterpenoids**	**MO**	**84.30**	**87.46**
Aliphatic sesquiterpenes Monocyclic sesquiterpenes Bi–and tricyclic sesquiterpenes	AS MS BS	2.03 0.26 0.32	1.5 0.34 0.36
**Sesquiterpenes**	**S**	**2.61**	**2.2**
Aliphatic sesquiterpenoids Monocyclic sesquiterpenoids Bi–and tricyclic sesquiterpenoids	ASO MSO BSO	0.51 0.71 0.47	0.50 0.40 0.37
**Sesquiterpenoids**	**SO**	**1.69**	**1.27**
**Others**	**OT**	**3.78**	**3.25**

Abbreviations: AM: aliphatic monoterpenes; MM: monocyclic monoterpenes; BM: bi-and tricyclic monoterpenes; AMO: aliphatic monoterpenoids; MMO: monocyclic monoterpenoids; BMO: bi-and tricyclic monoterpenoids; AS: aliphatic sesquiterpenes; MS: monocyclic sesquiterpenes; BS: bi- and tricyclic sesquiterpenes; ASO: aliphatic sesquiterpenoids; MSO: monocyclic sesquiterpenoids; BSO: bi- and tricyclic sesquiterpenoids; OT: others.

**Table 3 molecules-26-05317-t003:** Antioxidant activity of *L. angustifolia* essential oils.

Samples	IC_50_ ABTS (mg/mL)
*LA 2019*	1.00 ± 0.07
*LA 2020*	1.7 ± 0.1
Ascorbic acid (positive control)	0.032 ± 0.008

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
