# Peer review of "Chemical Profile, In Vitro Biological Activity and Comparison of Essential Oils from Fresh and Dried Flowers of Lavandula angustifolia L."

_molecules, 2021, doi:10.3390/molecules26175317_

Round 1

Reviewer 1 Report

The manuscript by Caprari et al. describes the chemical composition and biological activity of the essential oil from the flowers of Lavandula. A comparison between two batches has been also presented. The research has been carried out properly and the results are of interest.

Some minor points should be addressed before acceptance.

1) English language is poor. Many grammar mistakes and some poorly written sentences are present throughout the manuscript. See for example:

Title: "biological activity in vitro" should be changed in "in vitro biological activity"

Abstract: "antibacterial activity assay in vitro" should be changed in "in vitro antibacterial activity assay"

Page 1, line 32: "its" should be replaced by "their"

Page 2, line: 55: "shown" should be replaced by "showed that"

Page 2, line: 64: "pose" should be replaced by "poses"

Page 2, lines: 74 and 88: "will be" should be replaced by "was" or "were"

Page 2, lines: 79-80: "A previous authors" should be rephrased

Page 2, line: 82: get rid of comma

Page 3, line: 98: "antibacterial and the antifungal activities in vitro" should be changed in "in vitro antibacterial and the antifungal activities"

Page 9, line 211: "southern" should be changed in "Southern"

Page 9, line 212: "north" should be changed in "North"

Page 9, line 216: Please avoid the term "dose" and use "concentration", as this is not an animal study

page 12, line 273: "countries" should be changed in "regions"

page 12, line 280: get rid of dot before "[36]"

page 12, line 281: get rid of comma after "data"

page 13, lines 284-287: please rephrase

page 13, line 297: get rid of "of"

page 14, line 381: "µm" is not correctly displayed

page 14, line 391: "n-alkanes", "n" should be in italic

page 15, line 437: "are" shold be changed in "were"

page 16, line 488: get rid of the comma

2) The decimal figures in the tables should be consistent. Please change "11.00 ± 0.39", "35.31 ± 1.00", "15.67 ± 0.27", "6.51 ± 0.12", "9.00 ± 0.29", "36.03 ± 0.10", "19.35 ± 0.12" with "11.0 ± 0.4", "35 ± 1", "15.7 ± 0.3", "6.5 ± 0.1", "9.0 ± 0.3", "36.0 ± 0.1", "19.4 ± 0.1", respectively, in Table 1.

Please change "1 ± 0.072", "1.66 ± 0.138" with "1.00 ± 0.07", "1.7 ± 0.1", respectively, in Table 3.

3) Some chemical names containing symbols are not correctly displayed

4) Figure 4: Please change "1,8-Ceneole" within the figure

5) Figure 7: Please change "µl" with "µL"

Author Response

To Reviewer 1 for Molecules Journal

Object: Revised manuscript - 1339803 - Invitation for the Special Issue on “Biological Activity of Plant Compounds and Extractsbelongs to the Section Natural Products Chemistry.

Answers to review 1:

The authors express their gratitude for the timely and professional revision work carried out. We have taken the notes, suggestions and criticisms into account in the new, fully revised and highly scientifically improved version of the final manuscript.

Some minor points should be addressed before acceptance.

1) English language is poor. Many grammar mistakes and some poorly written sentences are present throughout the manuscript. We thank the Review. We submitted the manuscript to revision language by internal Service of Molecules – MDPI-English editing ID: english-33383

All of the following examples were changed.  See for example:

Title: "biological activity in vitro" should be changed in "in vitro biological activity" DONE

Abstract: "antibacterial activity assay in vitro" should be changed in "in vitro antibacterial activity assay" DONE

Page 1, line 32: "its" should be replaced by "their" DONE

Page 2, line: 55: "shown" should be replaced by "showed that" DONE

Page 2, line: 64: "pose" should be replaced by "poses" DONE (now line 66)

Page 2, lines: 74 and 88: "will be" should be replaced by "was" or "were" DONE (now lines 80 and 93).

Page 2, lines: 79-80: "A previous authors" should be rephrased WE CHANGED (now lines 84-86)

Page 2, line: 82: get rid of comma DONE

Page 3, line: 98: "antibacterial and the antifungal activities in vitro" should be changed in "in vitro antibacterial and the antifungal activities" DONE (now lines 102-104)

Page 9, line 211: "southern" should be changed in "Southern" DONE (line 223)

Page 9, line 212: "north" should be changed in "North" DONE (line 224)

Page 9, line 216: Please avoid the term "dose" and use "concentration", as this is not an animal study. WE CHANGED (now line 228)

page 12, line 273: "countries" should be changed in "regions" DONE (line 293)

page 12, line 280: get rid of dot before "[36]" DONE

page 12, line 281: get rid of comma after "data" DONE

page 13, lines 284-287: please rephrase. WE CHANGED (lines 291-321)

page 13, line 297: get rid of "of" DONE

page 14, line 381: "µm" is not correctly displayed DONE

page 14, line 391: "n-alkanes", "n" should be in italic DONE

page 15, line 437: "are" should be changed in "were" WE CHANGED

page 16, line 488: get rid of the comma. DONE

2) The decimal figures in the tables should be consistent. Please change "11.00 ± 0.39", "35.31 ± 1.00", "15.67 ± 0.27", "6.51 ± 0.12", "9.00 ± 0.29", "36.03 ± 0.10", "19.35 ± 0.12" with "11.0 ± 0.4", "35.3 ± 1.0", "15.7 ± 0.3", "6.5 ± 0.1", "9.0 ± 0.3", "36.0 ± 0.1", "19.4 ± 0.1", respectively, in Table 1. WE CHANGED.

Please change "1 ± 0.072", "1.66 ± 0.138" with "1.00 ± 0.07", "1.7 ± 0.1", respectively, in Table 3. WE CHANGED

3) Some chemical names containing symbols are not correctly displayed. WE CHANGED

4) Figure 4: Please change "1,8-Ceneole" within the figure. WE CHANGED

5) Figure 7: Please change "µl" with "µL". WE CHANGED

Thank you for your kind suggestion

Reviewer 2 Report

The authors of the manuscript titled "Chemical profile, biological activity in vitro and comparison of essential oil from fresh and dried flowers of Lavandula angustifolia L." report the isolation and activity study of essential oils from fresh and dried flowers of Lavandula angustifolia L. The body of work presented here is appropriate for Molecules however, it needs some major revisions before it can be considered for publication.

Points that need to be addressed.

  1. The authors have reported the chemical composition of essential oils extracted based on the GC-MS. Have any efforts went into the isolation of the individual components. Do authors have any inputs towards which of the components reported in Table 1 are responsible for the activity of these essential oils?
  2. The authors should enlarge figures 2 and 3 for better visibility.
  3. Please enlarge the X and Y axes of figure 4 for better visibility.
  4. In the case of antibacterial and antifungal testing, the authors only picked a single strain of each to perform the study. They need to include more strains of both bacteria and fungi (resistant and sensitive strains if possible) to definitively conclude the antibacterial and antifungal activities of these essential oils. Moreover, the authors need to include positive controls for figure 7 and figure 8 studies.
  5. The authors need to explore the cytotoxicity of these essential oils.
  6. In the conclusion, section the authors need to make it clear the difference between fresh and dried flowers and how the composition and activity are changed.

Author Response

To Reviewer 2 for  Molecules Journal

Object: Revised manuscript - 1339803 - Invitation for the Special Issue on “Biological Activity of Plant Compounds and Extractsbelongs to the Section Natural Products Chemistry.

Answers to review 2:

The authors express their gratitude for the timely and professional revision work carried out. We have taken the notes, suggestions and criticisms into account in the new, fully revised and highly scientifically improved version of the final manuscript.

Points that need to be addressed.

  1. The authors have reported the chemical composition of essential oils extracted based on the GC-MS. Have any efforts went into the isolation of the individual components. Do authors have any inputs towards which of the components reported in Table 1 are responsible for the activity of these essential oils?

Thank you for your kind suggestion. In Discussion part (page 13) from line 292 to line 322, we clarified:

i. the main components of both essential oils (dried and fresh flowers from L. angustifolia) probably responsible for the described biological activities;

ii. we highlighted the differences (qualitative and quantitative) in the composition of both EOs;

iii. we discussed the results obtained from the antibacterial and antifungal test and the relationship with the main components, taking into account the current literature on the topic.

2. The authors should enlarge figures 2 and 3 for better visibility. WE CHANGED

3. Please enlarge the X and Y axes of figure 4 for better visibility. WE CHANGED

4. In the case of antibacterial and antifungal testing, the authors only picked a single strain of each to perform the study. They need to include more strains of both bacteria and fungi (resistant and sensitive strains if possible) to definitively conclude the antibacterial and antifungal activities of these essential oils. Moreover, the authors need to include positive controls for figure 7 and figure 8 studies.

We added info in page 1, abstract (lines 70-73); Page 2, lines 102-104;

Thank you for your kind suggestion. In the text, now we remark these observations and we suggest several more tests for antimicrobial screening. More, we have included in the manuscript some experiments performed to test the antibacterial activity of lavender EO also against B. subtilis PY79, chosen as a representative species of Gram-positive bacteria. Accordingly, a new Figure 6b was added. (Page 9)

Pages 237-242, With reference to positive controls, more details in the text have provided for Fig. 8.

On the other hand, for antibacterial activity assays we used a positive control (represented by amoxicillin and clavulanic acid) only for the disk diffusion tests. In fact, experiments aimed at analyzing the bactericidal/bacteriostatic effect of a substance (time-kill curves) usually include only one control consisting in untreated bacteria (see, for example, Guimarães et al. - Molecules, 2019, 24, 2471). Anyway, we believe that the suggestion of the reviewer is relevant and we will use Escherichia coli DH5α as a positive control in future experiments/researches aimed at analyzing the effects of lavender oil against a broader spectrum of bacterial species.

5. The authors need to explore the cytotoxicity of these essential oils.

We agree and in the text, we include the role of cytotoxicity test. (Page 14, lines 374-376)

6. In the conclusion section the authors need to make it clear the difference between fresh and dried flowers and how the composition and activity are changed.

In Conclusion, we added more details to better clear the differences between fresh and dried flowers; more, we cited notes on how, the composition and activity changed (page 18, lines 525-534).

Thank you for your kind suggestion.

Round 2

Reviewer 2 Report

The authors have addressed comments from the reviewers. The manuscript is ready to be published in molecules.